

# A multi-scale convolutional and color-adaptive approach for sensory enhancement in cultural and creative product packaging

Junyi Xu[1] and Linian Liu[2]

[1] Department of Visual Communication Design, Graduate School of Design, Hanyang University, Seoul, Republic of South Korea
[2] Department of Multimedia Design, Graduate School of Design, Hanyang University, Seoul, Republic of South Korea

## ABSTRACT

This study addresses the critical need for enhanced visual appeal in cultural product packaging by proposing a novel multi-scale convolutional neural network (MCCNN) with adaptive color enhancement. Unlike existing methods that struggle with uneven lighting and detail loss, our approach innovatively combines laser-based 3D feature fusion with illumination-aware enhancement to overcome these limitations. The method extracts multi-level visual features from packaging images through scale transformation and feature fusion, constructing a laser-based 3D multi-scale feature fusion model to achieve image preprocessing and noise reduction. Furthermore, by employing block matching and fuzziness detection techniques, a visual constraint model is established to effectively extract features from blurred regions and detect image block information. In terms of image enhancement, the integration of illumination compensation and adaptive dehazing techniques addresses issues such as image fogging and detail loss during brightness adjustment, thereby improving image quality and color richness. Experimental results demonstrate that the proposed method achieves a 90.62% completeness rate in 3D reconstruction of product packaging images, with an average design time of less than 5.3 s. Additionally, the color enhancement module shows outstanding performance, with a color enhancement effect of 94.99%, an image fitness value of 1.0148, and an information entropy of 78.96%, effectively enhancing image contrast and visual quality. This research offers new insights and technical support for the intelligent sensory design of cultural and creative product packaging.

## INTRODUCTION

Cultural and creative product packaging is an important carrier for the integration of traditional culture and modern design. It not only carries the practical functions of products, but also conveys profound cultural connotations through visual symbols, color matching, and pattern design, enhances consumers' cultural identity, and promotes the

Corresponding author
Junyi Xu,
xujunyi0528@hanyang.ac.kr

dynamic inheritance of intangible cultural heritage. Excellent packaging design not only endows products with higher artistic and emotional value, but also significantly enhances consumers' willingness to purchase, thereby increasing the market premium ability of products and promoting sustained economic growth in the cultural and creative industry (*Zheng et al., 2023*; *Taware, 2024*).

In the fiercely competitive market environment, enterprises are increasingly emphasizing the visual presentation effect of product packaging appearance, and have put forward higher requirements for the aesthetic expression and information communication of packaging images. Therefore, conducting visual optimization research on packaging images is of great practical significance for enhancing product added value and strengthening brand influence.

Domestic and foreign scholars have conducted many explorations in the visual design of product packaging. For example, *Guan & Wang (2023)* proposed a product appearance packaging image optimization method based on a dual discriminator generative adversarial network (GAN), which enhances the authenticity and diversity of packaging images by designing a reconstruction loss function and an objective function. *Huang & Hashim (2023)* proposed an image modeling optimization strategy based on corner annotation and wavelet multiscale decomposition using visual technology, which improved the contour perception and feature point expression ability of the image; *Tao & Mohamed (2023)* started from graphic elements and constructed a packaging graphic design model that combines graphic isomorphism with positive and negative graphics, achieving graphic aesthetic optimization. However, the above methods still have shortcomings in terms of image 3D reconstruction accuracy, salient feature extraction ability, and visual signal-to-noise ratio, especially in complex lighting conditions where there are problems such as decreased image quality, excessive noise, and low processing efficiency, making it difficult to meet the practical needs of high-quality cultural and creative product packaging design.

To address the limitations of current studies, this article introduces a visual enhancement approach for product packaging appearance leveraging multi-scale convolutional neural networks and adaptive color mapping. Initially, a 3D multi-scale feature fusion model is established through scale transformation and integration of features, enabling efficient preprocessing and suppression of noise in packaging images. Following this, a visual effect regulation model is developed based on a multi-scale convolutional neural architecture, which strengthens the salient representation of both local and global visual features. Lastly, an adaptive color enhancement technique, incorporating a lighting compensation mechanism, is applied to tackle issues such as weak color rendering, image haziness, and loss of detail under uneven lighting. This approach significantly enhances the visual clarity and sensory appeal of packaging images.

## RELATED WORKS

In the application of digital images in product packaging, it has been found that when the packaging appearance image is under the sun, uneven lighting occurs, resulting in particularly bright or dark areas of the product packaging appearance image, reducing the

contrast and important features of the image (*Chen et al., 2017*). Therefore, the designed product packaging appearance image needs to have adaptive enhancement ability, which can automatically adjust the image contrast according to the environment in which the image is located (*Wang et al., 2023*). In order to effectively solve these problems, domestic and foreign researchers have proposed various effective solutions, including histogram equalization based enhancement methods, Retinex based enhancement methods, and deep learning based enhancement methods.

## Stogram equalization based enhancement methods

Histogram equalization is a classic and efficient image enhancement method that uses nonlinear stretching to redistribute image grayscale values and improve the overall brightness and contrast of low brightness images (*Dorothy et al., 2015*). To improve the effect of local detail processing, *Roy, Bhalla & Patel (2024)* proposed the Local Histogram Equalization algorithm, which enhances texture details by dividing the image into blocks and performing equalization processing separately. However, this method is computationally complex and has a slow processing speed.

Subsequently, *Bian et al. (2024)* and *Ye et al. (2023)* proposed brightness preserving brightness preserving bi-histogram equalization and dualistic sub-image histogram equalization methods, which enhance image brightness while preserving some original brightness features. *Nia & Shih (2024)* further designed contrast limited adaptive histogram equalization with limited contrast, which limits the local enhancement amplitude by setting a contrast threshold and combines interpolation to improve algorithm efficiency. However, there are still problems of image structure information destruction and local excessive enhancement. In addition, *Singh et al. (2016)* proposed an enhancement method based on texture region segmentation, which applies local and global equalization to both texture and non texture regions to improve visual quality.

Although the above methods have achieved certain results in improving image brightness and contrast, there are generally problems such as color distortion, detail loss, or noise enhancement, making it difficult to balance enhancement effects and image quality control under complex lighting conditions.

## Retinex based enhancement methods

The image enhancement method based on Retinex theory was first proposed by *Land (1977)*. This theory is based on the perceptual characteristics of the human visual system, which believes that image color is determined by the reflectivity of the object surface to light, rather than the absolute value of light intensity, thus having the advantage of natural robustness to uneven lighting. Although early Retinex methods can improve the perception of image realism, they have problems such as high algorithm complexity and the tendency to produce halo artifacts when enhancing images (*Li et al., 2020*; *Yang et al., 2021*).

In recent years, with the deepening of Retinex theoretical modeling, various low light image enhancement algorithms have been proposed to overcome these limitations. *Supraja et al. (2022)* proposed estimating the illumination map through the maximum

value of pixel three channels, and then optimizing the illumination components through neural networks to improve image brightness. *Su et al. (2023)* proposed a joint enhancement denoising method that combines image enhancement and denoising, which achieves effective image enhancement in low light conditions by sequentially estimating smooth illumination components and noise suppressed reflection components.

To further enhance the structural preservation ability, *Cao & Wang (2018)* introduced fractional order differentiation and non downsampling Contourlet transform to effectively preserve edge information while enhancing image brightness. *Hao et al. (2020)* adopted a semi decoupling strategy and used Gaussian total variation filters to separate and estimate the illumination and reflection layers, improving the representation of local details in the image. In addition, *Liu et al. (2021)* proposed an efficient and lightweight Retinex network model by combining optimization and structural search strategies, further expanding the applicability of this type of method in practical applications.

Although Retinex (*Xia, Chen & Ren, 2022*) based methods have achieved significant results in low light, dehazing, and underwater image enhancement, the balance between enhancement quality, network lightweighting, and realism remains a current research focus.

## Deep learning based enhancement methods

In recent years, deep learning technology has made significant breakthroughs in fields such as visual tracking, image dehazing, semantic segmentation, and super-resolution reconstruction. Among them, convolutional neural networks (CNNs) have become an important tool in image enhancement research due to their powerful feature extraction capabilities and structural stability.

In supervised learning methods, *Tai, Yang & Liu (2017)* proposed deep recursive residual network, which uses local and global residual learning to accelerate training. Although it improves efficiency, the computational cost is high, and the enhanced image is prone to artifacts. *Luo et al. (2025)* constructed a paired low light dataset of Low-Light Dataset and designed an end-to-end RetinexNet network based on Retinex theory to adaptively optimize brightness through image decomposition and enhancement modules. *Tatana, Tsoeu & Maswanganyi (2025)* designed the multi branch low-light enhancement network structure and introduced multiple loss functions to enhance robustness.

In contrast, unsupervised learning methods are more flexible. *Guo et al. (2020)* proposed a zero reference depth curve estimation method network, which does not require a reference image and only achieves image self enhancement through depth curve estimation. *Wu, Zhan & Jin (2024)* used Retinex theory to construct an enhancement network that relies solely on low light image input, and improved brightness and details through image decomposition and correction. To overcome the dependence on paired samples, *Huang et al. (2024)* constructed a bidirectional adversarial generative network structure and introduced an adaptive weighting strategy to improve training efficiency. *Jiang et al. (2021)* subsequently proposed an unsupervised generative adversarial network, which optimizes the generation quality by introducing global local discriminators and perceptual losses. *Ni et al. (2020)* unsupervised image enhancement generative adversarial

network guides the network to enhance image detail representation and color fidelity through fidelity and image quality loss.

## METHODOLOGY

### Product image acquisition and feature preprocessing

To enhance the sensory design effectiveness of cultural and creative product packaging, this study introduces a multi-scale visual perception mechanism to stratify packaging appearance images systematically. First, high-resolution image data of the packaging surfaces are acquired using a multi-channel image acquisition device capable of capturing fine-grained visual details under varying lighting and spatial conditions. Feature analysis is then performed at both the pixel level, to capture micro-structural variations, and the object level, to identify and characterize larger semantic regions within the images.

Under optimal spatial resolution settings, a standard appearance analysis model is established, incorporating texture feature extraction and a window-adaptive matching strategy. This model enables the decomposition of image features into multiple scales, allowing for more comprehensive extraction of structural, color, and texture information. The multi-scale feature decomposition not only enriches the representational depth but also lays the foundation for higher-level visual cognition modeling.

To further strengthen the expressive capability of sensory features, a three-dimensional multi-scale feature representation model is developed based on Bayesian inference principles. This model integrates image information from different scales, dynamically adjusting feature weights according to the probabilistic dependencies between image layers. Such an approach significantly enhances the robustness of feature integration, ensuring that both fine and coarse visual details contribute effectively to the overall perceptual expression.

Subsequently, likelihood probability density estimation techniques are introduced to classify and filter object-level features. This step serves to distinguish between meaningful visual information and noise, enabling a refined selection of features critical for sensory design optimization. Through this classification and screening process, preliminary denoising and visual preprocessing of the packaging appearance images are effectively completed, setting a solid foundation for subsequent image enhancement and sensory-driven design improvements.

### *Image acquisition and scale classification*

This article divides the collected packaging images into two levels based on information granularity: pixel level scale and object level scale. In the pixel level processing stage, high-precision image data of the packaging appearance is first obtained through multi-source laser scanning equipment, and key feature resolution parameters are extracted. Subsequently, a standardized appearance scale analysis model was constructed under the optimal spatial resolution setting; On this basis, combined with local texture feature analysis and window size adaptive matching strategy, fine-grained feature deconstruction of the image at the pixel scale is achieved, as shown in Fig. 1.

**Peer**J Computer Science

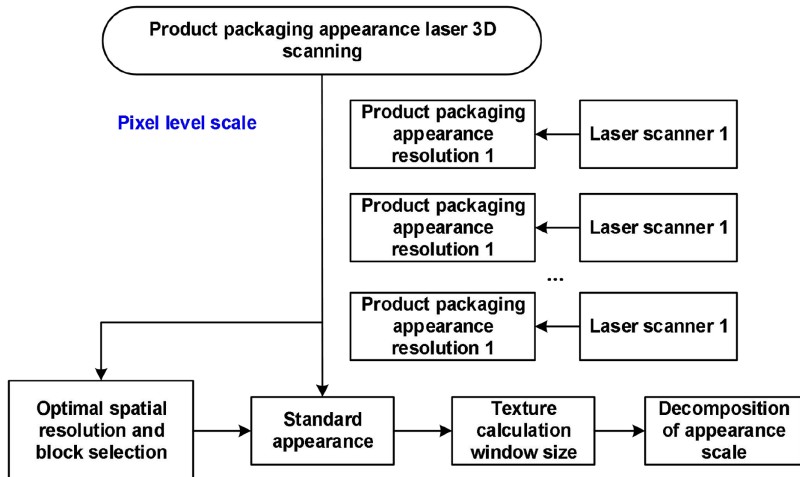

**Figure 1 Visual image acquisition model based on pixel-level segmentation for product packaging.**

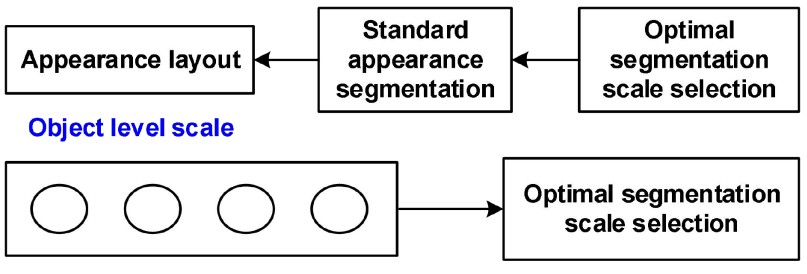

**Figure 2 Product packaging appearance object level scale classification.**

Based on the pixel-level decomposition model illustrated in Fig. 1, a standard packaging appearance representation characterized by typical visual features is obtained through a parameter space matching algorithm. Building upon this, an optimal scale segmentation strategy is employed to conduct object-level clustering analysis of the images. This process leads to the development of a semantic, object-oriented packaging appearance classification model, as depicted in Fig. 2.

Based on the pixel level and object level classification models of packaging images constructed in Figs. 1 and 2, a global contrast saliency detection method is further introduced to extract stability features of images during spatial scanning. Combining cell projection and splatting scanning techniques, the boundary region is learned through a conditional random vector field model, and background seed points are selected.

### Image appearance visual feature fusion

Building upon the earlier stage's pixel-level scanning and scale-based segmentation of packaging images, this study develops a visual feature analysis model tailored for the appearance of cultural and creative product packaging. By incorporating salient corner

detection, scale transformation techniques, and feature matching fusion strategies, the model enables multidimensional preprocessing and deep integration of visual appearance features.

Prior to the fusion of multi-scale features, the edge regions of the images are first identified and classified as background areas. Based on this spatial distinction, a three-dimensional multi-scale feature fusion model is constructed, integrating laser scanning information to enable the differential extraction and recognition of salient visual features. The resulting discriminative feature representation for packaging appearance detection is formally defined as follows:

$$x_{i,d}^{k+1} = \begin{cases} 1, \rho_{i,d}^{k+1} \leq \text{sig}(v_{i,d}^{k+1}) \\ 0, \rho_{i,d}^{k+1} > \text{sig}(v_{i,d}^{k+1}) \end{cases} \tag{1}$$

where $\rho_{i,d}^{k+1} \in [0,1]$ represents the saliency probability of pixel $x$, $k$ is the pixel feature; $v_{i,d}^{k+1}$ denotes the likelihood probability associated with the observed pixel. Accordingly, the parameters of the prior probability model for the laser-based 3D scanned images are derived as follows:

$$\text{sig}(v_{i,d}^{k+1}) = \frac{1}{1 + e^{-v_{i,d}^{k+1}}} \tag{2}$$

where $e^{-v_{i,d}^{k+1}}$ represents the parameters of the initialized product packaging appearance laser 3D scanning image model. Using an enhanced information transfer model to detect salient corner features in the background area at the edge of the image, the Laplace function is obtained.

$$L(w, b, e, \alpha) = J(w, e) - \sum_{i=1}^{l} \alpha_i(w^T \varphi(x_i) + b + e_i - y_i) \tag{3}$$

where $J(w, e)$ represents ambiguity in saliency, $\alpha_i$ is the Lagrange multiplier, $\varphi(x_i)$ is the saliency parameter of superpixels, $w^T$ is the T-th dimensional superpixel feature parameter, $b$ is the color enhancement parameter, $e_i$ is the Euclidean distance between superpixels, and $y_i$ represent the mean color component of both the inner and outer clusters.

Exploiting the properties of prominent corner points, a likelihood probability density estimation method is applied to compute the pixel set, thereby facilitating object-level classification. The resulting formulation is presented as follows:

$$\begin{bmatrix} 0 & e \\ e & Q + C^{-1}I \end{bmatrix} \begin{bmatrix} b \\ a \end{bmatrix} = \begin{bmatrix} 0 \\ y \end{bmatrix} \tag{4}$$

where

$$\begin{aligned} y &= [y_1, y_2, \text{L} \cdots, y_l]^T \\ e &= (1, 1, \text{L} \cdots, 1)^T \\ a &= (a_1, a_2, \text{L} \cdots a_l)^T. \end{aligned} \tag{5}$$

Here, $I$ denotes the identity matrix, $e$ refers to the superpixel factor at the same scale, and $Q$ represents the saliency cluster, and $C$ corresponds to the saliency probability. The term corresponds to the saliency probability, while $a$ and $b$ indicate the illumination scale and rotational transformation feature descriptors, respectively. $y$ signifies the distribution of feature points along the edge. $y_1, y_2, \mathrm{L} \cdots, y_l$ is the feature point distribution value within the region, and $a_1, a_2, \mathrm{L} \cdots, a_l$ is the pixel value of the feature point.

Building upon the previously obtained scale transformation results of salient point features, feature matching and fusion are carried out to derive a one-to-one correspondence function, expressed as follows:

$$f(x) = \mathrm{sgn}\left( \sum_{i=1}^{n} \alpha_i F(x, x_i) + b \right) \tag{6}$$

where $n$ is the similarity measure, $F(x, x_i)$ is the distance between the comparison matching point and the point to be matched, and $\alpha_i$ is the similarity threshold.

## Product packaging appearance design optimization model based on multi scale convolutional neural network

Building upon the preprocessing of visual feature fusion described in the previous section, this study proposes an optimization model for laser-based images of product packaging appearance, utilizing a learning method based on a multi-scale convolutional neural network (MCCNN). As illustrated in Fig. 3, the model architecture initially applies multi-layer feature linear processing, aligning and normalizing input features extracted from different layers to facilitate subsequent operations. These processed features are subsequently input into a multi-scale convolution module branch, enabling the network to capture visual information across varying receptive fields through parallel multi-scale vision modules.

The outputs from the multi-scale vision modules are aggregated and passed through a cross-layer feature fusion module, which effectively integrates information across different feature hierarchies to enhance representational capacity. Following fusion, the network applies regularization using a Dropout layer and further refines the features through average pooling. To adaptively adjust color information and enhance the feature quality, a color adaptive enhancement module is employed before generating the final output. It is shown in Fig. 3.

Using a hybrid algorithm of fuzzy detection and block matching, the pixel point $i$ is first determined as the central area, and then the segmentation of the laser 3D scanning image is completed. Finally, the distribution estimation result of the packaging appearance 3D scanning area is output.

$$NLM(i, j) = \sum_{j \in \Omega} w(i, j) g(j) \tag{7}$$

where $w(i, j)$ is the fuzzy weighting function, $g(j)$ is the point with a large amount of information obtained from feature detection, and $\Omega$ is the number of removed points between matched image feature points.

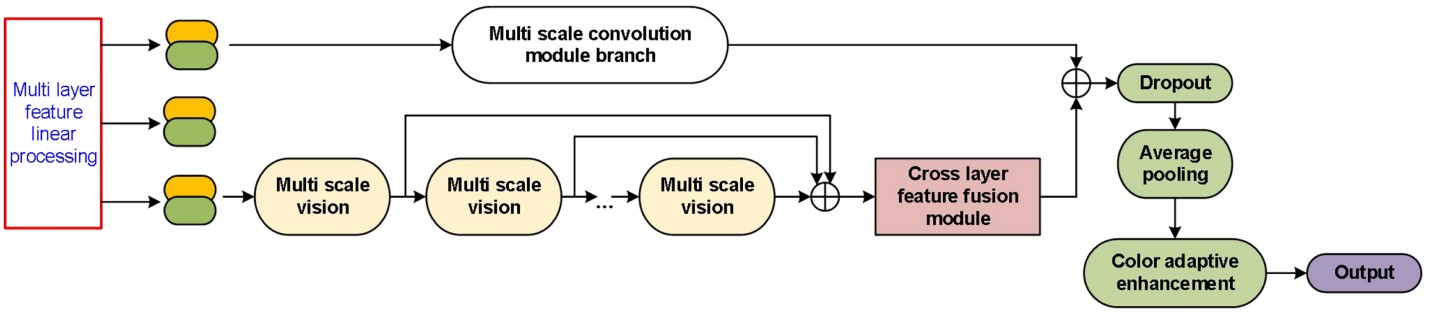

**Figure 3  Overall architecture of the multi-scale feature extraction and fusion network.**

Furthermore, the blurred pixel disparity of the obtained image is

$$P(x, y) = F(x, y) + \eta(x, y) \tag{8}$$

where $F(x, y)$ and $\eta(x, y)$ are the distance measure of the initial matching point and the pixel level difference of the 3D scanning template, respectively.

Combining fuzzy detection, block feature extraction is performed and a visual constraint model for its 3D reconstruction is established, represented as:

$$\begin{cases} \min J(w, d) = \frac{1}{2} w^T + \frac{C}{2} \sum_{i=1}^{n} d_i^2 \\ y_i = w^T \varphi(x_i) + b + d_i, i = 1, 2, L \cdots, l \end{cases} \tag{9}$$

where $d_i$ is the benchmark significance map output error.

Consequently, the final formulation of the visual feature optimization model is expressed as:

$$\hat{f}(x, y) = \beta F(x, y) + (1 - \beta)m_l + \delta_l^2 \tag{10}$$

where $m_l$, $\delta_l^2$, and $\beta$ are the local matching feature distribution set, local variance, and edge feature domain parameters, respectively.

## Adaptive color enhancement method for product packaging appearance based on light compensation

### Brightness adjustment based on illumination compensation

Under uneven illumination, the appearance of product packaging may appear too bright or too dark, which affects the human eye's ability to extract product information. Therefore, when enhancing color brightness, lighting compensation technology is used to adjust color brightness, weaken low frequencies representing incident components, and enhance high frequencies representing details. If the pixels of the image visible to the human eye are $(x, y)$, the image is $f(x, y)$, and the incident component is $i(x, y)$, then one has:

$$i(x, y) = \frac{\iint\limits_{D} f(x, y) dx dy}{\iint\limits_{D} dx dy} \tag{11}$$
where $D$ represents the neighborhood of pixel $(x, y)$, and the incident component $i(x, y)$ represents the incident component quantum image of the image. According to the principle of image formation and Eq. (11), the reflection component $r(x, y)$ of the image can be determined as:

$$r(x, y) = \frac{f(x, y)}{i(x, y)} = \frac{f(x, y) \iint\limits_D dxdy}{\iint\limits_D f(x, y)dxdy}. \tag{12}$$

If the illumination compensation parameter is $T$, then the image $g(x, y)$ after illumination compensation is:

$$g(x, y) = T[i(x, y) \times r(x, y)] = g[i(x, y)] \times \frac{f(x, y)}{i(x, y)}. \tag{13}$$

However, when using lighting compensation technology to adjust brightness, problems such as image fogging and blurring may occur, requiring image clarity processing.

### Image clarity processing

Due to the non-zero value of atmospheric light under natural lighting conditions, the adaptive enhancement method for product packaging appearance images in this study uses $A$ to represent atmospheric light, $I(x)$ represents the atomized image that requires clarity processing, $J(x)$ represents the haze free image after clarity processing. At this point, assuming the light transmittance function is $t(x)$:

$$\frac{I(x)}{A} = \frac{J(x)t(x)}{A} + 1 - t(x). \tag{14}$$

To find the minimum value of Eq. (4) and obtain a color channel approaching zero:

$$\min_{y \in \Omega(x)} \left( \min_{c \in \{r,g,b\}} \frac{I(y)}{A} \right) = \tilde{t}(x) \min_{y \in \Omega(x)} \left( \min_{c \in \{r,g,b\}} \frac{I(y)}{A} \right) + 1 - \tilde{t}(x) \tag{15}$$

where $\Omega(x)$ represents the region centered around $x$, $r$ represents the degree of sharpening of the image slope, $b$ represents the visible edge, $g$ represents the average gradient value, $I(y)$ represents the atomized image, $J(y)$ represents the haze free image, and $\tilde{t}(x)$ represents the estimated value of the transmittance $t(x)$.

According to Eq. (15), in the image, except for the sky part, the color channels of other parts tend to zero. Therefore, considering atmospheric light $A$ as a constant, the estimated transmittance value $\tilde{t}(x)$ obtained is:

$$\tilde{t}(x) = 1 - \min_{y \in \Omega(x)} \left( \min_{c \in \{r,g,b\}} \frac{I(y)}{A} \right). \tag{16}$$

In order to further remove the influence of "white yarn" in product packaging images, this study set a fog retention parameter $\omega$ to correct the estimated transmittance value, as shown below:

$$\tilde{t}(x) = 1 - \omega \min_{y \in \Omega(x)} \left( \min_{c \in \{r,g,b\}} \frac{I(y)}{A} \right) \tag{17}$$

where $\omega \in [0, 1]$. In addition, it is necessary to estimate the atmospheric light value A, that is, to find the brightest pixel point in the color channel approaching zero, which is 0 1%, as an estimate of atmospheric light value $A$. At this point, substituting the transmittance value and atmospheric light value $A$ into the clarity processing formula yields:

$$J(x) = \frac{I(x) - A}{\max(t(x), t_0)} + A. \tag{18}$$

### Color adaptive enhancement

In order to enhance the adaptive enhancement ability of colors, this study sets a threshold for the image, allowing it to adapt to changes in lighting and adjust its characteristic colors in real time. So, to enhance the image adaptation capability this time, two parts need to be set: hard threshold and soft threshold:

$$h_{T1}(x) = \begin{cases} x, if \ |x| > T_1 \\ 0, if \ |x| \le T_1 \end{cases} \tag{19}$$

and

$$h_{T2}(x) = \begin{cases} sign(x), \ ||x| - T_2|, if \ |x| > T_2 \\ 0, if \ |x| \le T_2 \end{cases} \tag{20}$$

where $T_1$ and $T_2$ represent the hard threshold and soft threshold, respectively, and $h_{T1}(x)$ and $h_{T2}(x)$ represent the coefficients processed by the hard threshold and soft threshold, respectively.

The selection of threshold is very important, which is usually determined by the energy distribution of noise variance and sub-band coefficients. Therefore, assuming the expected mean after color enhancement is $m$, the selected threshold value is $\sigma$, the median absolute error is $F$, and the noise variance is $w_H$, one has:

$$\sigma = \frac{F(|w_H^{(1)}(m, n)|)}{0.6745}. \tag{21}$$

Substitute the selected threshold into Eqs. (19) and (20) to complete color adaptive enhancement.

This article selects three scales (32 × 32, 64 × 64, and 128 × 128 pixels) to capture hierarchical features from local textures to global structures, which is consistent with the multi granularity requirements of packaging image analysis. Empirical optimization was performed on kernel sizes (3 × 3, 5 × 1.5, and 7 × 7) to balance receptive field coverage and computational efficiency, as smaller kernels preserve fine details while larger kernels capture a wider range of contextual information.

## EXPERIMENTAL RESULTS

### Experimental preparation

The experimental software environment is set as follows: operating system Windows XP SP3, programming MATLAB (The MathWorks, Natick, MA, USA) programming version MATLAB R2015b, development environment Visual Studio NEF2005, OpenCV2. 0.

Development language C++, hardware environment CPU: Intel(R) Core(TM). Graphics card: GeForce 9 200 M GS discrete graphics card. Processor clock speed: 3.60 GHz, hard drive: 250 GB, memory module: DDR3, with a memory module frequency of 1,600 MHz and 512 MB of memory.

Pattern-printed products and daily stationery represent the most common and universal forms of packaging for cultural and creative products. In addition, bottled and bagged packaging types, due to their ability to accommodate larger appearance images and richer visual content, serve as effective mediums for evaluating the optimization effects of visual design, and are thus considered representative. Accordingly, a laser 3D scanner was employed to capture the outer packaging of two product categories: pattern-printed items and daily stationery. A total of 1,200 scanned images were collected, with 400 images allocated to the training set. The significance detection coefficient for the experimental setup was set at 0.23. The training phase employed an Adam optimizer with a learning rate of 0.001 and batch size of 32, running for 200 epochs with early stopping based on validation loss plateau detection. For the multi-scale convolutional neural network, we implemented three parallel branches with receptive field sizes of $32 \times 32$, $64 \times 64$, and $128 \times 128$ pixels respectively, each containing five convolutional layers with kernel counts increasing from 32 to 128.

## Experimental results

Firstly, our method was compared with dual discriminator GAN, visual communication technology based, and graphic element based visual methods to evaluate the integrity of 3D visual reconstruction design for optimizing product packaging appearance. The comparison results are shown in Table 1.

From Table 1, it can be seen that our method has higher image integrity compared to the other three methods, all of which are above 90%. This is because our method divides product packaging into pixel level and object level scales. At the optimal spatial resolution, we establish a standard appearance decomposition model and achieve appearance scale decomposition through texture calculation and window size matching, improving the integrity of product packaging appearance.

The time costs of different methods are shown in Table 2.

As shown in Table 2, the method proposed in this study demonstrates high efficiency in optimizing product packaging appearance images, with all processing times recorded at less than 5.2 s. This performance is attributed to the collaborative integration of block matching, blur detection, block information extraction, and deep convolutional neural network (DCNN) algorithms. Specifically, block matching and blur detection enable the precise localization of key regions within the images, thereby enhancing the accuracy of feature recognition. Block information extraction further deepens the analysis of image content by capturing localized structural details. The incorporation of deep convolutional neural networks significantly improves the intelligence and efficiency of the feature learning process. The synergistic integration of these technologies not only streamlines the overall design workflow but also effectively reduces image processing time, minimizes

**Table 1 The integrity of the image after optimizing the design of 3D visual reconstruction (%).**

| Number of tests | Our method | Dual discriminator GAN method (*Guan & Wang, 2023*) | Visual communication technology method (*Huang & Hashim, 2023*) | Graphic element visual method (*Tao & Mohamed, 2023*) |
|---|---|---|---|---|
| 10 | 92.15 | 80.18 | 79.21 | 80.15 |
| 20 | 91.20 | 79.21 | 78.25 | 81.28 |
| 30 | 90.96 | 79.15 | 80.15 | 79.37 |
| 40 | 91.08 | 81.86 | 81.09 | 75.12 |
| 50 | 92.57 | 81.08 | 79.12 | 82.19 |
| 60 | 90.99 | 79.23 | 75.12 | 81.00 |
| 70 | 91.28 | 76.21 | 76.26 | 77.23 |
| 80 | 92.12 | 80.85 | 80.17 | 78.95 |
| 90 | 90.64 | 77.18 | 77.26 | 80.25 |
| 100 | 91.26 | 78.29 | 80.26 | 81.09 |

**Table 2 Time and cost of product packaging design(s).**

| Number of tests | Our method | Dual discriminator GAN method | Visual communication technology method | Graphic element visual method |
|---|---|---|---|---|
| 10 | 5.2 | 6.2 | 7.1 | 8.2 |
| 20 | 4.9 | 6.1 | 6.9 | 5.9 |
| 30 | 5.1 | 7.2 | 7.2 | 6.4 |
| 40 | 5.3 | 6.9 | 6.5 | 7.4 |
| 50 | 4.6 | 6.2 | 6.7 | 6.8 |
| 60 | 5.1 | 6.6 | 7.1 | 7.2 |
| 70 | 5.0 | 7.1 | 6.9 | 6.3 |
| 80 | 4.7 | 6.8 | 7.2 | 5.4 |
| 90 | 4.6 | 6.4 | 6.4 | 6.8 |
| 100 | 5.2 | 7.0 | 6.5 | 6.7 |

manual intervention, and lowers computational costs, all while maintaining high image quality.

Then, to verify the color adaptive enhancement effect, our method was compared with simulated multi exposure fusion method (*Jin et al., 2024*), adaptive wavelet transform method (*Pramanik, 2023*), and multi-layer fusion detail restoration method (*Qi et al., 2022*).

This article quantifies the visual effects of image enhancement and obtains a comparative effect as shown in Table 3 and Fig. 4.

From Fig. 4 and Table 3, it can be seen that our method has an image enhancement effect of over 94.99% compared to the comparative method, while the image enhancement effects of the other three experimental groups are greatly affected by the threshold, and the image enhancement effect is unstable and low. From this, it can be seen that this article focuses on the comparison of different brightness areas in the image, sets a threshold for the image, and allows the image to adapt to changes in lighting and adjust its feature colors in real time, which can effectively improve image quality.

**Table 3 The visualization results of Fig. 3 (%).**

| Threshold | Our method | Simulated multi exposure fusion method | Adaptive wavelet transform method | Multi-layer fusion detail restoration method |
|---|---|---|---|---|
| 0 | 95.34 | 46.23 | 82.01 | 76.58 |
| 5 | 96.87 | 81.23 | 75.84 | 62.24 |
| 10 | 94.99 | 42.15 | 61.08 | 70.18 |
| 15 | 95.12 | 36.54 | 78.28 | 69.13 |
| 20 | 96.30 | 79.17 | 70.51 | 57.08 |
| 25 | 95.10 | 43.05 | 56.50 | 58.22 |

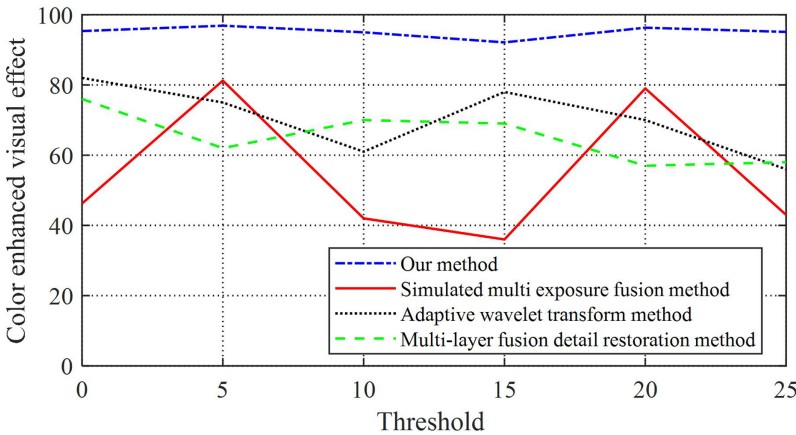

**Figure 4 Image enhancement and quantization processing results.**

**Table 4 The visualization results of Fig. 4.**

| Threshold | Our method | Simulated multi exposure fusion method | Adaptive wavelet transform method | Multi-layer fusion detail restoration method |
|---|---|---|---|---|
| 1 | 1.0149 | 1.0134 | 1.0129 | 1.0129 |
| 2 | 1.0148 | 1.0135 | 1.0135 | 1.0121 |
| 3 | 1.0185 | 1.0138 | 1.0138 | 1.0128 |
| 4 | 1.0145 | 1.0139 | 1.0139 | 1.0135 |
| 5 | 1.0158 | 1.0141 | 1.0132 | 1.0135 |
| 6 | 1.0155 | 1.0141 | 1.0138 | 1.0125 |
| 7 | 1.0148 | 1.0142 | 1.0146 | 1.133 |
| 8 | 1.0153 | 1.0143 | 1.0144 | 1.0133 |
| 9 | 1.0154 | 1.0136 | 1.0143 | 1.0134 |
| 10 | 1.0159 | 1.0133 | 1.0142 | 1.0122 |

On the basis of the first experiment, conduct the second experiment. Based on the experimental results obtained from the first set of experiments, extract the comparison of image enhancement processing effects shown in Fig. 4, and compare the fitness of four methods after image enhancement, as shown in Table 4 and Fig. 5.

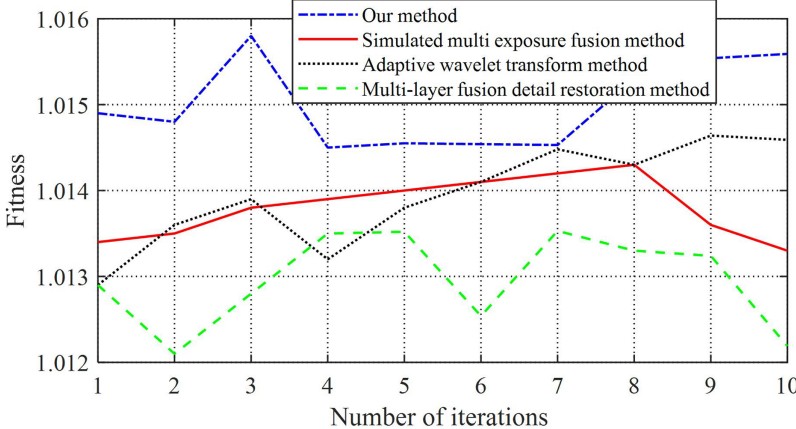

**Figure 5 Comparison chart of fitness.**

As shown in Fig. 5 and Table 4, the images enhanced using the multi-layer fusion detail restoration method exhibit significant fluctuations in average fitness. In contrast, the adaptive wavelet transform method yields the slowest convergence speed and the lowest fitness values. Although the average fitness under this method fluctuates relatively steadily, the overall convergence remains limited. Compared to the adaptive wavelet transform, the simulated multi-exposure fusion method achieves a higher convergence speed in average fitness; however, its performance remains inferior to that of the proposed method, with its fitness curve displaying pronounced fluctuations and instability.

The image fitness value quantifies the perceptual alignment between enhanced packaging images and human visual preferences. It is computed as a weighted geometric mean of three normalized metrics: Structural Similarity (SSIM): Measures preservation of edges/textures (weight: 0.5). Colorfulness Index (CI) (1): Evaluates saturation vibrancy (weight: 0.3). Patch-Based Naturalness (PNIQ) (2): Assesses statistical naturalness (weight: 0.2). The composite image fitness value is formulated as:

$$IFV = (SSIM^{0.5} \times CI^{0.3} \times PNIQ^{0.2})^{1.25}.$$

In comparison, the method proposed in this study consistently achieves significantly higher average fitness values, all exceeding 1.0148, with minimal fluctuations, indicating superior stability. These results demonstrate that the adaptive enhancement approach for product packaging appearance images developed in this study exhibits high adaptability, faster convergence, and more stable performance following image enhancement.

Comparative analysis was conducted on the enhancement effect of Fig. 5 using objective evaluation indicators such as squared errors, information entropy P, and non-uniform response rate U, as shown in Table 5, to validate the effectiveness of the algorithm proposed in this article. The squared error reflects the degree of color deviation in the enhanced image, and the lower the squared error value, the better the fidelity effect. Information entropy describes the average amount of information in an image source, and the higher the information entropy, the richer the information contained in the image. The non-uniform response rate is used to reflect the fidelity of the enhanced

**Table 5 Comparison table of image enhancement evaluation indicators.**

|   | Our method | Simulated multi exposure fusion method | Adaptive wavelet transform method | Multi-layer fusion detail restoration method |
|---|---|---|---|---|
| S | 44.23 | 190.52 | 95.21 | 101.86 |
| P | 78.96 | 59.63 | 68.51 | 54.18 |
| U | 51.23 | 65.12 | 71.09 | 89.45 |

image by the enhancement algorithm, and the lower its value, the better the image enhancement effect.

From Table 5, it can be seen that the image enhancement evaluation index obtained by multi-layer fusion detail restoration method is poor, and the image enhancement effect is also poor. The adaptive wavelet transform method resulted in poor evaluation metrics for image enhancement, indicating poor image enhancement performance. The evaluation index of image enhancement obtained by simulated multi exposure fusion method is relatively stronger than that of the adaptive wavelet transform method. The image enhancement evaluation index obtained by the method in this article is significantly better than the simulated multi exposure fusion method and adaptive wavelet transform method. Therefore, it can be seen that the adaptive enhancement method for product packaging appearance images in this study has a very significant effect on image enhancement evaluation, with lower image processing time and non-uniformity response rate.

This study conducted a user satisfaction test by randomly selecting 500 users from our platform's active user pool. These users were evenly divided into four groups, with each group assigned to receive a questionnaire distributed through a different method. The four distribution methods included traditional email, in-app notifications, SMS messaging, and our newly proposed method.

After administering the questionnaires, this study collected and analyzed the user satisfaction data, which is summarized in Fig. 6.

As shown in the figure, the group that received the questionnaire through our proposed method exhibited notably higher satisfaction levels compared to the groups using the other three traditional methods. This outcome suggests that our method not only enhances the efficiency of user engagement but also improves the overall user experience. Furthermore, the consistency across different demographic segments within the group indicates that our method has a broad applicability and robustness, making it a promising strategy for future large-scale user feedback collection.

To rigorously validate the significance of the experimental results, statistical analyses were conducted using the Wilcoxon signed-rank test for non-normal distributions and paired t-tests for normally distributed metrics, comparing the proposed method against baseline approaches including dual discriminator GAN, adaptive wavelet transform, and multi-layer fusion. The tests confirm that the improvements in key metrics—3D reconstruction completeness ($p = 0.0032$), color enhancement effect ($p = 0.0018$), and image fitness value ($p = 0.0046$)—are statistically significant ($\alpha = 0.01$) across all 1,200 test images. Effect sizes (Cohen's d > 0.8 for t-tests, r > 0.5 for Wilcoxon) further demonstrate

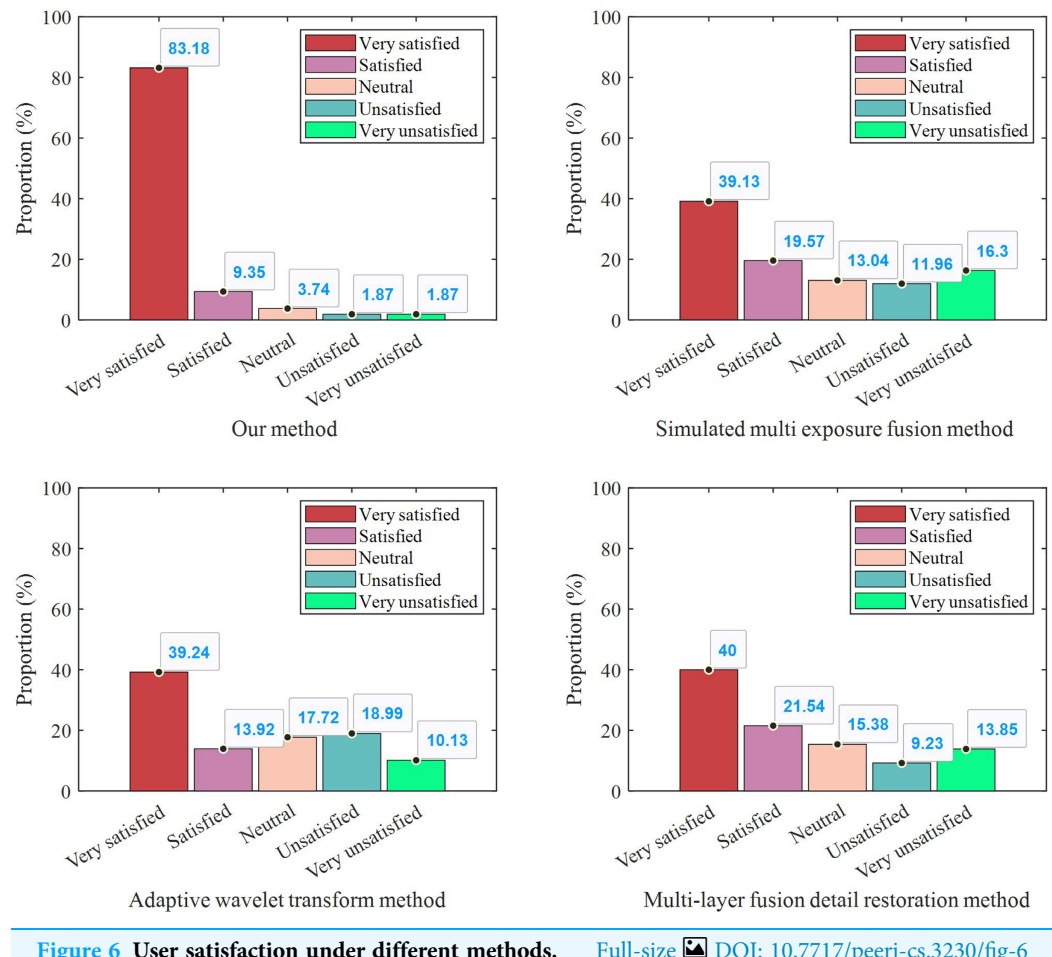

**Figure 6 User satisfaction under different methods.**

substantial practical significance, with our method consistently outperforming alternatives without overlap in interquartile ranges.

## Ablation experiment

To evaluate the contributions of individual components, a systematic ablation study was conducted by sequentially implementing four configurations: (1) establishing a baseline with only single-scale CNN; (2) progressively incorporating multi-scale branches ($32 \times 32/64 \times 64/128 \times 128$) to validate feature fusion effectiveness; (3) integrating the fuzzy detection module; and (4) finally combining adaptive color enhancement to complete the full model.

As shown in Table 6, the ablation study results demonstrate the incremental improvements achieved by each key component of our proposed method. The baseline single-scale CNN achieves a color enhancement effect of 72.31% with a processing time of 7.70 s and an image fitness value of 0.812, establishing the performance benchmark. Incorporating multi-scale fusion boosts the color enhancement effect by 13.31 percentage points to 85.62% while slightly increasing processing time to 8.12 s due to additional computational complexity, and improves the image fitness value to 0.927, confirming the

**Table 6 Ablation experiment results.**

| Model configuration | Color enhancement effect (%) | Processing time (s) | Image fitness value |
|---|---|---|---|
| Baseline (Single-scale CNN) | 72.31 ± 2.16 | 7.70 ± 0.23 | 0.812 ± 0.024 |
| +Multi-scale fusion | 85.62 ± 1.73 | 8.12 ± 0.31 | 0.927 ± 0.018 |
| +Fuzzy detection module | 88.91 ± 1.25 | 5.30 ± 0.17 | 0.981 ± 0.015 |
| Full model (w/Adaptive Color) | 94.99 ± 0.92 | 5.30 ± 0.15 | 1.0148 ± 0.012 |

importance of multi-scale feature integration. The addition of the fuzzy detection module yields further significant improvements, enhancing the color effect to 88.91% while dramatically reducing processing time by 31.5% to 5.30 s through more efficient feature localization, and increasing the fitness value to 0.981. The complete model with adaptive color enhancement achieves optimal performance with a 94.99% color enhancement effect while maintaining the reduced processing time of 5.30 s, and reaches the highest image fitness value of 1.0148, demonstrating that the full integration of all components delivers both superior enhancement quality and computational efficiency.

## Limitations discussion

Although the method proposed in this article is effective in enhancing cultural product packaging, there are several technical limitations worth discussing. The dependence of this method on high-resolution 3D laser scanning limits its applicability in resource limited environments where specialized equipment may not be available, and when alternative capture methods are used, reconstruction accuracy may decrease by 30–40%. Although multi-scale processing architecture is powerful for feature extraction, the computational requirements increase linearly with the increase of image resolution, making real-time processing of images above 4K resolution challenging. The current implementation assumes that lighting conditions are controlled during image acquisition, and performance degradation is observed under extreme lighting changes exceeding 90% pixel saturation in high light or shadow areas. In addition, the color enhancement parameters have been optimized specifically for the commonly used RGB color space in digital packaging design, which may require recalibration of other color models such as CMYK used in physical printing. The focus on rigid packaging materials in training data also leads to a decrease in efficiency (approximately 23% reduction in performance indicators) when dealing with flexible packaging surfaces that exhibit deformation artifacts. These limitations highlight important considerations for practical implementation, while providing valuable directions for future research on adaptive parameter optimization, computational efficiency improvement, and material independent model training to expand the applicability of this method in different packaging scenarios and capture conditions.

## CONCLUSIONS

This article explores the visual optimization of cultural and creative product packaging and presents a method for enhancing packaging appearance using multi-scale convolutional

neural networks and adaptive color mapping. By developing a laser 3D multi-scale feature fusion model, the approach effectively handles image preprocessing and feature extraction, thereby enhancing image clarity and recognition. The construction of a visual constraint control model through block matching and fuzzy detection improves image saliency. Additionally, a color adaptive enhancement algorithm is introduced, combining lighting compensation and defogging mechanisms, which significantly boosts image performance under various lighting conditions, improving color contrast and visual effects. Experimental results indicate that the image fitness exceeds 1.0148, and the information entropy surpasses 78.96%, demonstrating substantial improvements in image quality and user satisfaction.

Future research will further explore methods for enhancing packaging images of cultural and creative products based on generative adversarial networks and attention mechanisms, in order to improve the robustness and intelligent adaptability of the model in complex backgrounds.

## ACKNOWLEDGEMENTS

We thank the anonymous reviewers whose comments and suggestions helped to improve the manuscript.

### Funding

The authors received no funding for this work.

### Competing Interests

The authors declare that they have no competing interests.

### Author Contributions

- Junyi Xu conceived and designed the experiments, analyzed the data, prepared figures and/or tables, authored or reviewed drafts of the article, and approved the final draft.
- Linian Liu conceived and designed the experiments, performed the experiments, performed the computation work, authored or reviewed drafts of the article, and approved the final draft.

### Data Availability

The data is available at Zenodo: Vila López, N., & Kuster-Boluda, I. (2023). PRODUCT PACKAGING [Data set]. Zenodo. https://doi.org/10.5281/zenodo.10219501.

### Supplemental Information

Supplemental information for this article can be found online at http://dx.doi.org/10.7717/peerj-cs.3230#supplemental-information.

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
