# Peer review of "A multi-scale convolutional and color-adaptive approach for sensory enhancement in cultural and creative product packaging"

_PeerJ Computer Science, doi:10.7717/peerj-cs.3230_

## Round 0.1 · original submission · Major Revisions

Dear authors

Thanks for your submission, after careful consideration and experts comments, I'm here with sending you the reviewers comments for incorporation and improvements. Therefore, please carefully revise the paper in light of these comments and submit a detailed point by point response.

Thanks

·

Basic reporting

This paper presents a technically sound and innovative approach to enhancing the visual quality of cultural and creative product packaging using multi-scale convolutional neural networks and adaptive color enhancement techniques. The authors effectively combine feature extraction, noise reduction, and image enhancement methods—including illumination compensation and dehazing—to improve image clarity and color richness. The methodology is well-articulated, with a clear research objective and robust experimental validation, including impressive metrics such as a 90.62% 3D reconstruction completeness rate and a 94.99% color enhancement effect. The manuscript is generally well-written and structured, though minor improvements in language clarity and figure labeling could enhance readability. Overall, the study offers valuable contributions to the field of intelligent packaging design and demonstrates strong potential for practical application.

Experimental design

The authors effectively combine feature extraction, noise reduction, and image enhancement methods—including illumination compensation and dehazing—to improve image clarity and color richness.

Validity of the findings

The methodology is well-articulated, with a clear research objective and robust experimental validation, including impressive metrics such as a 90.62% 3D reconstruction completeness rate and a 94.99% color enhancement effect.

Additional comments

The manuscript is generally well-written and structured, though minor improvements in language clarity and figure labeling could enhance readability. Overall, the study offers valuable contributions to the field of intelligent packaging design and demonstrates strong potential for practical application.

Reviewer 2 ·

Basic reporting

This paper proposes a multi-scale CNN-based method combined with adaptive color enhancement for optimizing the visual quality of cultural and creative product packaging. While the approach demonstrates promising results, the paper exhibits several methodological and structural weaknesses that must be addressed.
The paper introduces a Multi-Scale Convolutional Neural Network (MCCNN) architecture but does not sufficiently justify the specific choices made (e.g., number of scales, kernel sizes, fusion strategy). A comparative ablation study exploring alternative architectures (e.g., UNet, DenseNet) is necessary to validate the model's design.
The model heavily integrates standard image processing techniques such as block matching and
Retinex-based compensation, yet it claims to be a deep learning approach. The paper should clarify the novelty of integrating these techniques with CNNs and explain why this combination is more effective than using GANs or transformer-based models for image enhancement.
Several equations presented in Section 3.2 (e.g., saliency probability, feature matching fusion) are insufficiently defined, with many symbols (e.g.,

Experimental design

The use of Windows XP and MATLAB R2015b raises concerns about compatibility, scalability, and
reproducibility. It is strongly recommended to migrate the implementation to a contemporary deep
learning framework (e.g., PyTorch, TensorFlow) and provide code access.
The “image fitness value” (1.0148) is repeatedly cited but not well-defined in the paper. The
authors must explain how it is computed, what it measures, and why it is a reliable metric for visual
packaging enhancement.

Validity of the findings

Although multiple metrics are provided across several methods, no statistical analysis (e.g., t-test,
Wilcoxon signed-rank test) is used to validate whether the observed improvements are significant.

Additional comments

N/A

Annotated reviews are not available for download in order to protect the identity of reviewers who chose to remain anonymous.

Reviewer 3 ·

Basic reporting

The introduction should give a fuller explanation of the research background, why the study is important, and what it aims to achieve. The structure of the paper can be improved by organizing the sections more clearly and making the figures and tables easier to understand. It is also important to clearly mention the hypotheses, as they help explain the purpose of the study and how the results should be understood. Adding screenshots of the experiment or user interface would make the study more transparent and easier to follow for others.

Experimental design

no comment

Validity of the findings

no comment

Additional comments

no comment

Annotated reviews are not available for download in order to protect the identity of reviewers who chose to remain anonymous.

Reviewer 4 ·

Basic reporting

The authors propose a sensory design method for cultural and creative product packaging based on multi-scale convolutional neural networks and adaptive colour enhancement. The method extracts multi-level visual features from packaging images through scale transformation and feature fusion, constructing a laser-based 3D multi-scale feature fusion model to achieve image pre-processing and noise reduction. Furthermore, a visual constraint model is established to effectively extract features from blurred regions and detect image block information by employing block matching and fuzziness detection techniques. The proposed approach seems to enhance the visual clarity and sensory appeal of packaging images significantly. The paper seems interesting and promising results are presented. However, the following should be addressed for improvement.

1. Keywords should be written according to a single format (considering alphabetical order, etc.).
2. “Error! Reference source not found.” should be corrected in the tables.
3. The Abstract must be self-contained and concisely describe the reason for the work, motivation, novelty or originality, methodology, results, and conclusions. The background analysis of the abstract is not enough.
4. Equations should be correctly written. Many of them are distorted and seem confusing. Necessary references should also be supplied.
5. Equations are not cited in the paper. They should be mentioned with the correct equation numbers. Do not use “…given as…”, “…we have…”, etc. Many of the equations are part of the sentences and that is why special attention is needed for correct sentence formation.
6. Some of the variables in the equations listed in the definitions need to be explained. Some mathematical notations are not rigorous enough to correctly understand the contents of the paper. The authors are requested to recheck all the definition of variables and further clarify these equations. Definitions of all variables and their intervals should be given.
7. All variables should be written in italic as in the equations.
8. Referencing within the text should be used using a single journal format.
9. Blank character should be correctly used. See for example: “…image[5].”, “equation(15),”, etc.
10. Pay special attention on the usage of abbreviations. Please write the explanation in full the first time an acronym or abbreviation is used. Then proceed with the shortened version.
11. Are the simulation results taken from the equal conditions? There is not any discussion. Add further details on how simulations were conducted. Similarly, resource and system characteristics could be added to Tables for clarity.
12. Technical challenges and limitations of these proposed methods should be explained. Clarifying the study’s limitations allows the readers to better understand under which conditions the results should be interpreted. A clear description of limitations of a study also shows that the researcher has a holistic understanding of his/her study.
13. The analysis and configurations of experiments should be presented in detail for reproducibility. A table with parameter setting for experimental results and analysis should be included in order to clearly describe them. The values for the parameters of the algorithms selected for comparison should be given.

Experimental design

.

Validity of the findings

.

---

## Round 0.2 · Minor Revisions

Dear authors,

Thank you for your submission after revising the paper. The majority of the reviewers are satisfied with the revised version; however, there are a few suggestions for improvement. Therefore, we invited you to update and resubmit.

Reviewer 2 ·

Basic reporting

The study offers a thorough summary of its goals, methods, and findings and is well-written and organized. The main issue with the visual limitations of cultural product packaging is successfully identified, and a Multi-Scale Convolutional Neural Network (MCCNN) combined with adaptive color enhancement is suggested as a novel solution. The study provides a solid justification for the suggested strategy by highlighting the shortcomings of current approaches and drawing on pertinent earlier research.

Experimental design

The suggested approach was contrasted with current feature extraction and image enhancement methods. To demonstrate advancements in color accuracy, computational efficiency, and visual clarity, both qualitative and quantitative comparisons were made.

Validity of the findings

Strong experimental protocols, suitable evaluation metrics, and insightful comparative analyses all support the study's findings, which exhibit strong internal and external validity.
Important factors like lighting, image clarity, and feature distribution that could affect the result are successfully controlled for in the experimental design. The reliability of the results is increased by using a diverse dataset that includes images with different kinds of degradation (such as fog, blur, and noise). The study makes sure that improvements in image quality and reconstruction are caused by the suggested method and not by outside influences by combining laser based 3D feature fusion and adaptive enhancement techniques within a multi-scale CNN framework.
The assessment is further guaranteed by the use of objective, measurable metrics, such as 3D reconstruction completeness (90.62%), image fitness (1.0148), and information entropy (78.96%).

Reviewer 3 ·

Basic reporting

N/A

Experimental design

N/A

Validity of the findings

N/A

Additional comments

See my comments in the attached file.

Annotated reviews are not available for download in order to protect the identity of reviewers who chose to remain anonymous.

Reviewer 4 ·

Basic reporting

The authors have not addressed many of my previous concerns. For example:

1. Blank character should be correctly used. See for example: “…image[5].”, “equation(15),”, etc.
2. Referencing within the text should be used using a single journal format. Different referensing styles are used.
3. Some of the variables in the equations listed in the definitions need to be explained. Some mathematical notations are not rigorous enough to correctly understand the contents of the paper. The authors are requested to recheck all the definition of variables and further clarify these equations. Definitions of all variables and their intervals should be given.
4. Equations should be correctly written. Many of them are distorted and seem confusing. Necessary references should also be supplied. Please check the final pdf of your revised paper.
5. Equations are not cited in the paper. They should be mentioned with the correct equation numbers. Do not use “…given as…”, “…we have…”, etc.
6. Many of the equations are part of the sentences and that is why special attention is needed for correct sentence formation.
7. “Error! Reference source not found.” should be corrected in the tables. See for example: Table 1.

Furthermore followings should be addressed:

1. "Structural Similarity (SSIM): Measures preservation of edges/textures (weight: 0.5). Colorfulness Index (CI) [1]: Evaluates saturation vibrancy (weight: 0.3). Patch Based Naturalness (PNIQ) [2]: Assesses statistical naturalness (weight: 0.2). The composite image fitness value is formulated as" should be corrected. Are [1] and [2] references?
2. "Table 3 The visualization results of Figure.3 (%)" should be corrected.

Experimental design

.

Validity of the findings

.

Additional comments

.

---

## Round 0.3 · accepted · Accept

Thank you for your update and submission. After the assessment of the revised paper, we are pleased to inform you that your manuscript is being recommended for publication.

Thank you for your fine contribution.